# Innate Lymphocyte Th1 and Th17 Responses in Elderly Hospitalised Patients with Infection and Sepsis

**DOI:** 10.3390/vaccines8020311

**Published:** 2020-06-17

**Authors:** John Davis Coakley, Eamon P. Breen, Ana Moreno-Olivera, Alhanouf I. Al-Harbi, Ashanty M. Melo, Brian O’Connell, Ross McManus, Derek G. Doherty, Thomas Ryan

**Affiliations:** 1St James’s Hospital Intensive Care Unit, James’s Street, Dublin 8, Ireland; thomasryan1958@gmail.com; 2Trinity Translational Medicine Institute, St James’s Hospital, Dublin 8, Ireland; BREENEA@tcd.ie; 3Department of Immunology, Trinity Translational Medicine Institute, Dublin 8, Ireland; ana.moreno.olivera@gmail.com (A.M.-O.); aalharbi@tcd.ie (A.I.A.-H.); ashanty.melorz@udlap.mx (A.M.M.); dohertde@tcd.ie (D.G.D.); 4Department of Clinical Microbiology, St James’s Hospital, James’s Street, Dublin 8, Ireland; BOConnell@stjames.ie; 5Molecular Medicine, Trinity Translational Medicine Institute, Department of Clinical Medicine, Trinity Centre for Health Sciences, St James’s Hospital, Dublin 8, Ireland; RMCMANUS@tcd.ie

**Keywords:** sepsis, infection, human, immunity, lymphocyte, innate, natural killer, gamma delta, MAIT, Th17

## Abstract

*Background:* the role of innate immunity in human sepsis must be fully clarified to identify potential avenues for novel immune adjuvant sepsis therapies. *Methods:* A prospective observational study was performed including patients with sepsis (septic group), infection without sepsis (infection group), and healthy controls (control group) in the setting of acute medical wards and intensive care units in a 1000-bed university hospital. A total of 42 patients with sepsis, 30 patients with infection, and 30 healthy controls were studied. The differentiation states of circulating mucosal associated invariant T (MAIT) cells and Natural Killer T (NKT) cells were characterised as naive (CD45RA^+^, CD197^+^), central memory (CD45RA^−^, CD197^+^), effector memory (CD45RA^−^, CD197^−^), or terminally differentiated (CD45RA^+^, CD197^−^). The differentiation states of circulating gamma-delta T lymphocytes were characterised as naive (CD45RA^+^, CD27^+^), central memory (CD45RA^−^, CD27^+^), effector memory (CD45RA^−^, CD27^−^), or terminally differentiated (CD45RA^+^, CD27^−^). The expression of IL-12 and IL-23 receptors, the transcription factors T-Bet and RORγt, and interferon-γ and IL-17a were analysed. *Results:* MAIT cell counts were lower in the septic group (*p* = 0.002) and the infection group (*p* < 0.001) than in the control group. The MAIT cell T-Bet expression in the infection group was greater than in the septic group (*p* = 0.012). The MAIT RORγt expression in the septic group was lower than in the control group (*p* = 0.003). The NK cell counts differed in the three groups (*p* < 0.001), with lower Natural Killer (NK) cell counts in the septic group (*p* < 0.001) and in the infection group (*p* = 0.001) than in the control group. The NK cell counts increased in the septic group in the 3 weeks following the onset of sepsis (*p* = 0.028). In lymphocyte stimulation experiments, fewer NK cells expressed T-Bet in the septic group than in the infection group (*p* = 0.002), and fewer NK cells expressed IFN-γ in the septic group than in the control group (*p* = 0.002). The NKT cell counts were lower in the septic group than both the control group (*p* = 0.05) and the infection group (*p* = 0.04). Fewer NKT cells expressed T-Bet in the septic group than in the infection group (*p* = 0.004). Fewer NKT cells expressed RORγt in the septic group than in the control group (*p* = 0.003). Fewer NKT cells expressed IFN-γ in the septic group than in both the control group (*p* = 0.002) and the infection group (*p* = 0.036). *Conclusion:* The clinical presentation of infection and or sepsis in patients is linked with a mosaic of changes in the innate lymphocyte Th1 and Th17 phenotypes. The manipulation of the innate lymphocyte phenotype offers a potential avenue for immune modulation in patients with sepsis.

## 1. Introduction

Although sepsis accounts for more fatalities than many common cancers and is particularly lethal in the elderly [1], the pathophysiology of sepsis in humans is yet to be clearly defined. Animal models portraying sepsis as a cytokine storm do not accurately reflect the human sepsis pathophysiology [2]. Patients with sepsis predominantly demonstrate a failure of innate immunity and adaptive immunity [3,4]. Human sepsis is characterised by impaired pathogen elimination, increased apoptosis, the reduced production of proinflammatory cytokines by a broad range of lymphocytes, and an emergence of regulatory T cells [5,6,7,8,9,10]. As a consequence, immunosuppression has recently become recognised as an obvious pivotal role in the pathophysiology of human sepsis [11].

We have recently shown that human sepsis is linked with a delayed failure to activate CD3^+^CD4^+^ lymphocytes to a Th17 phenotype [12]. Innate lymphocytes play pivotal roles in T cell polarisation via the rapid secretion of Th1, Th2, Th17, and regulatory T(Treg) cytokines. In the presence of such a profound paresis of adaptive immunity, innate lymphocytes may assume a crucial role in the pathophysiology of human sepsis, and their role in generating host immunity should be clarified in order to identify potential avenues for novel immune adjuvant therapies and also to provide an enhanced suite of biomarkers of innate immunity in human sepsis.

Innate lymphocytes and T cells with innate properties such as mucosal associated invariant T (MAIT) cells and Natural Killer T (NKT) cells bridge the gap between innate and adaptive immunity [13]. MAIT cells are the most numerous invariant T lymphocytes in the peripheral circulation and are also found in the gastrointestinal and respiratory tracts [14,15]. MAIT cells are activated by the vitamin B-related products of bacterial metabolism, bound to the major histocompatibility complex (MHC)-1-like antigen-presenting molecule MR-1, and express cytokines such as interferon-γ (IFN-γ) and interleukin-17 (IL-17). Reduced MAIT cell numbers have been linked with pseudomonal infections in cystic fibrosis patients and with streptococcal infection in patients with septic shock [16,17]. However, it is unclear whether the altered MAIT cell counts and phenotypes found in patients with infection are similar to those found in patients with overt sepsis.

Other innate lymphocytes, such as Natural Killer (NK) cells, Natural Killer T (NKT) cells, and gamma-delta T (γδ T) cells also have strong influence on the Th1/Th2/Th17/Treg balance of cytokine production. Whereas NK cells are stimulated through a variety of activating and inhibitory receptors, NKT cells respond to bacterial and autologous glycolipid antigens presented by CD1d. Both NK cells and NKT cells are important sources of early IFN-γ but can also be induced to produce other Th cytokines such as IL-17 [18,19,20], and these cells are variably reported as reduced or increased in patients with sepsis [21,22]. Currently, little is known of the role of the NKT cell phenotype in human sepsis. γδ T cells are also important innate sources of Th1 (IFN-γ and TNF-α), Th2 (IL-4 and IL-13), Th17 (IL-17), and Treg (IL-10) cytokines. Human γδ T cells can be divided into three main subsets. The most abundant subset found in blood, Vδ2 T cells, recognise pyrophosphate antigens produced by some bacteria bound to butryophilin 3A1, whereas Vδ1 and Vδ3 T cells can recognise glycolipids presented by CD1 and stress-inducible proteins found in tumour and virus-infected cells [23,24,25].

We used flow cytometry to examine the numbers, phenotypes, and activation status of circulating innate lymphocyte populations from healthy controls and patients with sepsis. We also investigated the potential of these cells to produce Th1 and Th17 cytokines by examining the IL-12 receptor and IL-23 receptor expression, respectively, and by measuring the ability of these cells to express the Th1 and Th17-associated transcription factors, RORγt and T-Bet, and produce IL-17 and IFN-γ following stimulation in vitro. In our study, we included an additional group of patients from the wards with infection. These patients did not have sepsis. We included this group to compare a normal immune response to infection with the immune response in sepsis, rather than just comparing to a control group with an immune system not challenged by infection. Given the prevalence of sepsis in the elderly and immune cellular senescence in elderly patients [26], this study focused on older hospitalised patients.

## 2. Methods and Materials

### 2.1. Study Population

Ethical approval was granted by the St. James’s hospital Research and Ethics Committee (Ethic approval code 2015-03). Informed written consent was obtained from patients. When this was not possible, we obtained assent from the next of kin, as is permitted by the National Consent Advisory Group for the ethical conduct of research [27]. We collected samples of blood, clinical data, and appropriate information. Within the septic group (42 patients), there were 32 patients assigned to the phenotype group and 10 patients assigned to the stimulation group. Within the infection group (30 patients), there were 20 patients assigned to the phenotype group and 10 patients assigned to the stimulation group. Within the control group (30 age matched healthy donors), there were 20 donors assigned to the phenotype group and 10 donors assigned to the stimulation group. We conducted this study between 2016 and 2018.

Septic group: The Sepsis 3 definitions were used to identify patients with sepsis in the St James’s Hospital intensive care unit [28]. The septic patients were then screened with the inclusion and exclusion criteria. If they met the inclusion criteria and consent was obtained, we then drew blood samples. These samples were obtained in the first 72 h of admission. This was time point “0”. We then drew 3 more blood samples every 7 days unless the patient was discharged from hospital or died. These were time points “1”, “2”, and “3”.

Infection group: The microbiology service identified patients with proven clinical infection but not sepsis. We recruited these patients from hospital wards in St James’s Hospital. We drew blood within 72 h of a clinically significant positive culture. We also drew a second sample of blood 7 days later, unless the patient had been discharged or died. A patient in the phenotype infection group was removed, reducing the number to 19; this was due to hyperbilirubinemia causing uninterpretable flow cytometry.

Control group: Donors for this group were obtained from the community. The donors did not have any infective symptoms in the previous 8 weeks. We drew blood at one time point for this group.

Samples for the cell stimulation were only taken at the time point “0”. The patient demographics (Table 1) have been reported in a previous study [12]. Our exclusion criteria are detailed in Table 2.

### 2.2. Immune Phenotyping of Circulating Lymphocytes

Freshly drawn whole blood was used for this analysis. Firstly, it was stained with a dead cell stain. The blood was then incubated with fluorochrome-conjugated monoclonal antibodies (mAb). These mAbs were used to identify cell types, differentiation status, and receptor expression. A BD FACS^TM^ lysing solution was then used to lyse the erythrocytes. The samples were then acquired using a FACSCanto^TM^ II flow cytometer (BD Biosciences). The flow cytometry analysis was performed using Treestar’s FlowJo^TM^ v10.4.2 software. Gating was conducted on lymphocytes, and analysis was performed once the doublets and dead cells were excluded. The compensation parameters were set using single stained controls. The gates were set using fluorescence-minus-one controls. Full blood counts were used to determine the cell counts.

The mAbs used were CD3 (BW264/56), CD56 (AF12-7H3, REA196), CD45RA (REA562), CD197 (CCR7; REA546), CD27 (M-T271, REA499), IL-12Rβ2 (REA333) and IL-23R (218213), Anti-TCR-Vδ1 (REA173), Anti-TCR-Vδ2 (123R3, REA771), CD161 (REA631, 191B8), Anti-TCR-Vα7.2 (REA179), and CD8 (BW135/80, REA734) (R&D Systems, Abingdon, UK and Miltenyi Biotec, Gladbach Bergische, Germany). The dead cells stain used was LIVE/DEAD Fixable Aqua dead cell stain, Molecular Probes, Leiden, The Netherlands.

The NK cells were identified as CD3^−^CD56^+^. The differentiation states of the circulating MAIT (CD3^+^CD161^+^Vα7.2^+^) and NKT (CD3^+^CD56^+^) cells were characterised as naive (CD45RA^+^, CD197^+^), central memory (CD45RA^−^, CD197^+^), effector memory (CD45RA^−^, CD197^−^), or terminally differentiated (CD45RA^+^, CD197^−^). The differentiation states of the circulating γδ T cell lymphocytes were characterised as naive (CD45RA^+^, CD27^+^), central memory (CD45RA^−^, CD27^+^), effector memory (CD45RA^−^, CD27^−^), or terminally differentiated (CD45RA^+^, CD27^−^).

The details of the experiment protocols can be found on the following open access link: dx.doi.org/10.17504/protocols.io.bfvhjn36.

### 2.3. Lymphocyte Stimulation In Vitro

Freshly drawn blood was prepared by density gradient centrifugation over Lymphoprep^TM^ (Axis-Shield, Dundee, UK) to produce peripheral blood mononuclear cells (PBMCs). The stimulation of 0.5 million cells was performed for 5 h using 50 ng/mL of phorbol myristate acetate with 1 μg/mL of ionomycin (PMA/I). Brefeldin-A was added to the wells intended for the intracellular staining of IFN-γ and IL-17A. After stimulation, a dead cell stain was added. The cell surface markers were then labelled with fluorochrome-conjugated mAbs. The fixation and permeabilization of the cells were performed with 4% paraformaldehyde and 0.2% saponin, respectively. The cells were then stained intracellularly with fluorochrome-conjugated mAbs specific for IFN-γ and IL-17a. The FoxP3 Staining Buffer Set (Miltenyi Biotec) was used in place of the paraformaldehyde and saponin above for the intra-cellular staining of T-Bet and RORγt. A cell fixation was performed again once they were stained. An immediate acquisition of the samples was performed. The acquisition and analysis were as per the immune phenotyping above in Section 2.2.

The mAbs used were as above in Section 2.2, and mAbs IFN-γ (REA600), IL17A (CZ8-23G1), T-Bet (REA102), and RORγt (REA278). The dead cell stain used was LIVE/DEAD Fixable Near IR dead cell stain, Thermo Fisher Scientific, Massachusetts, US. The protocols for these experiments can be found at the following open access links: dx.doi.org/10.17504/protocols.io.bfvgjn3w and dx.doi.org/10.17504/protocols.io.bfvfjn3n.

For an analysis with flow cytometry, the gating strategy was as per Section 2.2 and an example of this can be seen in Figure 1.

### 2.4. Statistical Analysis

The SPSS^®^ Statistical Software (IBM Corp, NY, USA) program was used for the analysis of the statistics. Wilcoxon/Kruskal Wallis testing was employed to compare the differences between the three patient groups (represented by a capped line in the figures). The *p* values for subsequent comparisons between the individual groups were Bonferoni adjusted for multiple comparisons (represented as a n-zigzag line in the figures). Chi Square testing was employed to compare the categorical variables.

A mixed effects general linear regression model was employed to analyse repeated measurements. Where a significant change in a repeated measurement was detected, the repeated assays were compared with the initial assay values with Bonferoni adjusted *p* values for multiple comparisons.

Throughout the analysis, a *p* value of less than 0.05 was considered significant.

## 3. Results

### 3.1. Demographics

Table 1 outlines the demographic data. The three groups had similar age demographics. There were more male than female patients in the immunophenotyping study. This is consistent with sepsis being a disorder of the elderly and more so in males [29]. The septic group had higher organ failure scores (*p* < 0.0001) and Apache II scores (*p* < 0.0001) than the infection group. In the immune phenotyping study, 13 (40%) patients in the septic group died, whereas mortality was 0% in the control and infection groups. The phenotype of the cells presented in this paper are those at the first time point unless otherwise stated.

### 3.2. MAIT Cells

The percentage of MAIT cells among the lymphocytes in peripheral circulation differed across the three patient groups (*p* = 0.03), with the percentage of MAIT cells being lower in the infection group compared with in the control group (*p* = 0.03) (Figure 2B). The percentage of MAIT cells expressing CD8 was similar in the three groups (Figure 2C). The absolute MAIT cell counts differed across the three groups (*p* < 0.001), with MAIT cell counts being lower in the septic group (*p* = 0.002) and the infection group (*p* < 0.001) than in the control group (Figure 2D). In the septic group, the MAIT cell counts did not change over time (Figure 2E). Thus, the MAIT cells appear to be depleted in both sepsis and infection.

The percentages of naive MAIT cells differed across the three groups (*p* < 0.001) and were lower in the infection group (*p* < 0.001) and the septic group (*p* = 0.003) than in the control group (Figure 2F). Similarly, the percentages of central memory MAIT cells were different between the groups (*p* = 0.029), with lower percentages of central memory MAIT cells in the septic group than in the control group (*p* = 0.051) (Figure 2F). The percentages of effector memory and terminally differentiated MAIT cells were similar in the three groups. The decrease in the naive and central memory MAIT cells in patients with sepsis suggests that MAIT cells are activated both in patients with sepsis and with infection.

The percentage of MAIT cells expressing IL-12Rβ2 and IL-23R was similar in the three patient groups, with IL-23R expression being more prevalent than that of IL-12Rβ2 (Figure 3A,B). The absolute numbers of IL-23R-expressing MAIT cells differed across the patient groups (*p* = 0.004), with the MAIT IL-23R^+^ cell counts being lower in the septic group (*p* = 0.003) than in the control group (Figure 3C). The MAIT IL-23R^+^ cell count in the septic group did not change over time (Figure 3D). The discordance between the percentage and absolute counts of MAIT IL-23R^+^ cells in patients may arise from relative lymphopoenia and redistribution into extra vascular compartments in patients with sepsis.

In the absence of stimulation, the T-Bet expression in MAIT cells differed across the three patient groups (*p* = 0.015), with T-Bet expression being greater in the infection group than in the septic group (*p* = 0.012) (Figure 4A). When the MAIT cells were stimulated with phorbol myristate acetate and ionomycin (PMA/I), the T-Bet expression was different in the three groups (*p* = 0.008), with the T-Bet expression being greater in the infection group than in the septic group (*p* = 0.006) (Figure 4A). In the absence of stimulation, the terminally differentiated MAIT cell expression of T-Bet was different in the three groups (*p* = 0.007), with T-Bet expression being greater in the infection group than in the septic group (*p* = 0.006) (Figure 4C). Thus, the MAIT cells expressed a predominant Th1 phenotype in patients with infection but not in those with sepsis.

When MAIT cells were stimulated with PMA/I, the RORγt expression was different in the three groups (*p* = 0.004), with the RORγt expression being lower in the septic group than in the control group (*p* = 0.003) (Figure 4B). The RORγt expression in effector memory MAIT cells stimulated with PMA/I differed in the three groups (*p* = 0.008), with the RORγt expression being lower in the septic group than in the control group (*p* = 0.008) (Figure 4D). Thus, the MAIT cells of septic patients failed to express a Th17 phenotype, which is consistent with the depletion of IL-23R expression in MAIT cells in patients with sepsis. MAIT cells stimulated with PMA/I exhibited a similar expression of IFN-γ and IL-17a in the three patient groups (Figure 4E). Thus, the clinical presentation of infection or sepsis is linked both with MAIT cell counts but also with the MAIT cell phenotype.

### 3.3. CD3^+^ CD161^+^ Lymphocytes

CD3^+^ lymphocytes expressing CD161^+^ include a composite population of lymphocytes capable of secreting IL-17, including both adaptive and innate lymphocytes such as MAIT and NKT cells [30,31].

The percentage of CD3^+^CD161^+^ lymphocytes was similar in the three patient groups (Figure 5A). There was a significant difference in the percentages of naive CD3^+^ CD161^+^ lymphocytes in the three patient groups (*p* = 0.001); the percentages of naive CD3^+^CD161^+^ lymphocytes were lower in the infection group than either in the control group (*p* = 0.001) or the sepsis group (*p* = 0.004) (Figure 5B). There was a significant difference in the percentages of central memory CD3^+^CD161^+^ lymphocytes in the three groups (*p* = 0.001), with percentages of central memory CD3^+^CD161^+^ lymphocytes being lower in the infection group than in either the control group (*p* = 0.002) or the septic group (*p* = 0.012) (Figure 5B). The percentages of effector memory and terminally differentiated CD3^+^CD161^+^ lymphocytes were similar in all the patient groups (Figure 5B). The decrease in naive and central memory CD3^+^CD161^+^ lymphocytes in patients with infection reflects the activation of these lymphocytes, which was not present in patients with sepsis. As many of these CD3^+^CD161^+^ lymphocytes are adaptive and thus MHC-restricted lymphocytes, the CD3^+^CD161^+^ lymphocyte activation failure in patients with sepsis likely reflects the well-described downregulation of antigen presentation mechanisms in sepsis. This contrasts with innate non-MHC-restricted innate lymphocytes, such as MAIT cells, that are activated equally in patients with sepsis and infection.

With regard to absolute counts, CD3^+^CD161^+^ lymphocytes increased in patients with sepsis from 106 +/− 15 (mean +/− standard error) to 211 +/− 64 (mean +/− standard error) over the course of the study (*p* = 0.047) (Figure 5C). The absolute number of effector memory CD3^+^ CD161^+^ lymphocytes (*p* = 0.006) and terminally differentiated CD3^+^CD161^+^ lymphocytes (*p* = 0.004) increased in the septic group in the three weeks after admission (Figure 5C).

In the unstimulated CD3^+^CD161^+^ lymphocytes, the T-Bet expression differed across the three groups (*p* = 0.025), with a greater T-Bet expression in the infection group than in the septic group (*p* = 0.02) (Figure 5D). In unstimulated CD3^+^CD161^+^ lymphocytes, the RORγt expression differed across the three groups (*p* = 0.004), with a lower RORγt expression in the septic group than in the control group (*p* = 0.003) (Figure 5D). The IL-17A and IFN-γ expression was similar in the CD3^+^CD161^+^ lymphocytes from the three patient groups when stimulated with PMA/I (Figure 5D). Thus, in a similar manner to MAIT cells, CD3^+^CD161^+^ lymphocytes from patients with infection exhibit an augmented Th1 response, whereas the CD3^+^CD161^+^ lymphocytes of patients with sepsis exhibit a deficient Th17 response.

Nine of the patients with sepsis died. Among the patients with sepsis, the percentage of all the lymphocytes that were CD3^+^ CD161^+^ lymphocytes was lower in patients who died (*n* = 9, median 3%, 95% confidence interval 1–7%) versus in survivors (*n* = 23, median 9%, 95% confidence interval 7–17%) (Mann U Whitney test, *p* = 0.008) (Figure 5E).

### 3.4. Natural Killer (NK) Cells

The percentage of NK cells within the pool of circulating lymphocytes was different across the three groups (*p* = 0.002), with proportionately fewer NK cells in the septic group than in the infection group (*p* = 0.02) and the control group (*p* = 0.006) (Figure 6B). In the septic group, the percentage of NK cells increased over the four study times (*p* = 0.004), from 5% (+/−0.7%) to 11.1% (+/−1.7) (Figure 6D).

The NK cell counts differed in the three groups (*p* < 0.001), with lower NK counts in the septic group (*p* < 0.001) and the infection group (*p* = 0.001) than in the control group (Figure 6C). The NK cell counts increased in septic patients over the four study time points (*p* = 0.028) (Figure 6E).

The majority of NK cells were CD56^dim^ (NK Dim) cells. The expression of IL-23R in NK cells was similar across the three patient groups (Figure 7A). The expression of IL-12Rβ2 in NK cells differed significantly across the three groups (*p* = 0.008), with the NK cell IL-12Rβ2 expression lower in the septic group than in the control group (*p* = 0.05) and lower in the infection group than in the control group (*p* = 0.01) (Figure 7A). The number of NK IL-12Rβ2 lymphocytes differed across the three groups (*p* = 0.002), with fewer NK IL-12Rβ2 lymphocytes in the septic group (*p* = 0.006) and the infection group (*p* = 0.003) than in the control group (Figure 7B). However, the expression of IL-12Rβ2 by NK cells was so small that it is unlikely to be of clinical significance.

The NK cells stimulated with PMA/I showed that the percentage of NK cells expressing T-Bet differed in the three groups (*p* = 0.002), with fewer lymphocytes expressing T-Bet in the septic group than in the infection group (*p* = 0.002) (Figure 7C). The NK cells stimulated with PMA/I also showed that the percentage of NK cells expressing IFN-γ was different in the three groups (*p* = 0.003), with fewer NK cells of patients with sepsis expressing IFN-γ than the controls (*p* = 0.002) (Figure 7D). Thus, the septic patients exhibited a deficient NK Th1 response.

### 3.5. Natural Killer T (NKT) Cells

The percentages of NKT cells differed across the three patient groups (*p* = 0.003), with more NKT cells in the infection group than in the septic group (*p* = 0.002) (Figure 8A). The percentage of NKT naive cells differed across patient groups (*p* < 0.001), with fewer NKT naive cells in the septic group (*p* = 0.01) and the infection group (*p* < 0.001) than in the control group (Figure 8E).

The NKT cell counts differed across the three groups (*p* = 0.008), with lower NKT cell numbers in the septic group than in the control group (*p* = 0.04) and the infection group (*p* = 0.02) (Figure 8B). The NKT cell counts did not change over time in the septic patients (Figure 8c).

The naive NKT cell counts differed across the three groups (*p* < 0.001), with fewer naive NKT cells in the septic group than in the control group (*p* < 0.001) and fewer in the infection group than in the control group (*p* = 0.001) (Figure 8F). There were scant NKT central memory cells in the three groups. The NKT effector memory cell counts were different in the three groups (*p* = 0.02), with more NKT effector memory cells in the infection group than in the septic group (*p* = 0.035) (Figure 8F). The NKT terminally differentiated cells differed in the three groups (*p* = 0.002), with fewer NKT terminally differentiated cells in the septic group than in the control group (*p* = 0.03) and the infection group (*p* = 0.003) (Figure 8F). These lower naive NKT cell counts and increased NKT effector memory cells reflect NKT cell activation in sepsis and infection.

The NKT IL-23 receptor (IL-23R) expression was similar across the three patient groups (Figure 8d). However, the expression of IL-12 receptor (IL-12Rβ2) differed significantly across the three groups (*p* = 0.004), with the IL-12Rβ2 expression lower in the infection group than in the control group (*p* = 0.003) (Figure 8D). So few NKT cells expressed either IL-12Rβ2 or IL-23R that the comparisons of the cell counts between groups were deemed irrelevant.

Without stimulation, the percentage of NKT cells expressing T-Bet differed in the three groups (*p* = 0.01), with fewer NKT cells expressing T-Bet in the septic group than in the infection group (*p* = 0.01) (Figure 9A). In stimulated NKT cells, the expression of T-Bet was different in the three groups (*p* = 0.005), with fewer NKT cells expressing T-Bet in the septic group than those in the infection group (*p* = 0.004) (Figure 9A). In the stimulated NKT cells, the expression of RORγt was different in the three groups (*p* = 0.003), with fewer NKT cells expressing RORγt in the septic group than in the control group (*p* = 0.003) (Figure 9B).

In the stimulated NKT cells, the expression of IFN-γ was different in the three groups (*p* = 0.002), with fewer NKT cells expressing IFN-γ in the septic group than in the control group (*p* = 0.002) and the infection group (*p* = 0.036) (Figure 9C). When stimulated, the NKT cells expressed similar IL-17A in the three groups. (Figure 9D). Thus, the septic patients exhibited deficiencies in both NKT Th1 and Th17 responses.

### 3.6. Gamma-Delta (γδ) T lymphocytes

The percentages of Vδ1 T cells were similar in the three groups (Figure 10C). However, the percentage of naive Vδ1 T cells were significantly different in the three groups (*p* = 0.001), with fewer naive Vδ1 T cells in the infection group than in the control group (*p* = 0.001) and fewer in the septic group than in the control group (*p* = 0.01) (Figure 10E). The absolute counts of Vδ1 T cells were similar in the three groups (Figure 10D).

The absolute counts of naive Vδ1 T cells were different in the three groups (*p* = 0.002); the naive Vδ1 T cell counts were lower in the septic group (*p* = 0.002) and the infection group (*p* = 0.016) than in the control group (Figure 10F). This data indicates Vδ1 T cell activation in both sepsis and infection, as was evident with the other non-MHC-restricted lymphocytes.

Few Vδ1 T cells expressed IL-12Rβ2 (Figure 10B). The expression of IL-23R was more frequent than that of IL-12Rβ2 and was significantly different across the three groups (*p* < 0.001) (Figure 10B). The Vδ1 IL-23R^+^ expression was different in the three groups (*p* = 0.001), with fewer Vδ1 T cells expressing IL-23R in the infection group than in the control group (*p* = 0.001) (Figure 10B).

The Vδ1 T cell expression of T-Bet and RORγt was similar in the three patient groups (Figure 11A,B). In Vδ1 T cells stimulated with PMA/I, the IFN-γ expression differed between the patient groups (*p* = 0.02), with a greater IFN-γ expression in the infection group than in the septic group (*p* = 0.028) (Figure 11C).

The percentage of Vδ2 T cells was significantly different in the three groups (*p* = 0.035), with fewer Vδ2 T cells in the septic group than in the control group (*p* = 0.034) (Figure 12A). The percentage of naive Vδ2 T cells was significantly different in the three groups (*p* < 0.001), with a lower percentage naive Vδ2 T cells in the infection group (*p* = 0.01) and the septic group (*p* < 0.001) than in the control group (Figure 12C). This again indicates the activation of Vδ2 T cells with both infection and sepsis.

The absolute number of Vδ2 T cells was different in the three groups (*p* < 0.001); the number of Vδ2 T cells was lower in the septic group (*p* < 0.001) and the infection group (*p* = 0.05) than in the control group (Figure 12B). The number of naive Vδ2 T cells was different in the three groups (*p* < 0.001), with fewer naive Vδ2 T cells in the septic group (*p* < 0.001) and the infection group (*p* = 0.001) than in the control group (Figure 12D). Similarly, the number of central memory Vδ2 T cells was different in the three groups (*p* = 0.004), with fewer central memory Vδ2 T cells in the septic group (*p* = 0.004) and the infection group (*p* = 0.03) than in the control group (Figure 12D). The number of terminally differentiated Vδ2 T cells was different in the three groups (*p* = 0.008), with fewer terminally differentiated Vδ2 T cells in the septic group (*p* = 0.03) and the infection group (*p* = 0.014) than in the control group (Figure 12D). Again, this indicates Vδ2 T cell activation in both infection and sepsis, even though the number of Vδ2 T cells appears to be depleted, which may reflect lymphocyte redistribution away from the intravascular space. Very few Vδ2 T cells expressed either IL-12Rβ2 or IL-23R in the three patient groups.

In stimulated Vδ2 T cells, the T-Bet expression differed across the three patient groups (*p* = 0.031), with a lower T-Bet expression in patients with sepsis than in the controls (*p* = 0.029) (Figure 13A). In unstimulated Vδ2 T cells, the RORγt expression differed across the three patient groups (*p* = 0.027), with a lower RORγt expression in patients with sepsis than in the controls (*p* = 0.054) (Figure 13B). In Vδ2 T cell stimulation with PMA/I, there was no difference in the IFN-γ expression between patient groups (Figure 13C). The expression of IL17A was not detectable in the stimulation experiment of Vδ2 T cells.

Within patients with sepsis, there was no link between mortality and γδ T cell, NK cell, or NKT cell percentages or receptor expressions.

## 4. Discussion

This study clearly links a mosaic of innate lymphocyte phenotypes with the clinical presentation of infection and sepsis in patients. Human sepsis studies of host immunity are problematic, as there is difficulty distinguishing between causal and coincidental molecular events. The very nature of human sepsis, with an unpredictable and often fulminant onset, precludes any attempt to establish the baseline immune functionality, which might predict subsequent sepsis onset or severity. To circumvent this problem, we included a third patient group in the study design. This third group of patients with infection, but without sepsis, may provide a more appropriate benchmark immune response from patients who tolerate infection with relative impunity rather than using healthy controls.

Prior studies have examined the association between innate lymphocytes and sepsis [17]. Grimaldi reported a decrease in MAIT cell count in patients with severe sepsis, but did not comment on the Th1/Th17 phenotype or distinguish between the MAIT cell responses to infection and sepsis. In contrast, the present study noted a decrease in circulating MAIT cells with infection in the absence of sepsis, which suggests that the numeric depletion of MAIT cells is a common feature of all infections and may reflect redistribution rather than depletion. Although there was evidence of MAIT cell activation in both sepsis and infection, sepsis was linked with an altered MAIT phenotype, as Th17 MAIT cells were reduced in sepsis, suggesting that the MAIT cell phenotype may be as important as absolute cell numbers in sepsis pathophysiology. As previously described by Venet, γδ T cells were decreased in patients with septic shock [32].

This association is biologically plausible, as immunity is promoted by Th17 cytokines against extracellular bacteria and fungi. They induce neutrophil recruitment to the site of inflammation by promoting antimicrobial peptide release and maintaining the intestinal barrier function [33,34]. In addition, the archetypic Th1 cytokine IFN-γ appears to be pivotally involved in generating an appropriate polymorphonuclear leukocyte bactericidal response and enhances antigen presentation by monocytes and macrophages, which in turn will give rise to an adaptive immune response [35].

It is notable that, despite the widespread expression of phenotype-specific transcription factors, few if any circulating lymphocytes expressed IL-12Rβ2 or IL-23R, with a considerably greater expression of IL-23R than IL-12Rβ2. Thus, circulating lymphocytes in patients with infection and sepsis may not as yet be fully activated, and hence assaying IL-17 cytokines from serum in patients with sepsis and infection may prove unrewarding [36]. However, the reduced expression of the Th17-specific transcription factor RORγt in lymphocyte populations of septic patients provides more robust evidence for the link between human sepsis and deficient Th17 responses in the majority of innate lymphocytes. Similarly, the enhanced expression of T-Bet by the innate lymphocytes of patients with infection but not sepsis provides inferential evidence that human sepsis is linked with a failure to elaborate a robust Th1 response in specific innate lymphocytes. While Carvelli recently reported a decrease in TH17 innate lymphoid cells in patients with sepsis, Carvelli did not include patients with infection, and could not link patient outcome to a specific cellular phenotype [37].

CD3^+^ CD161^+^ lymphocytes are a composite population of diverse IL-17-producing lymphocytes [31,38]. These CD3^+^ CD161^+^ lymphocytes are potent mediators of inflammation, and have been linked with chronic inflammatory diseases such as rheumatoid arthritis and correlate with disease severity in rheumatoid arthritis [39,40]. However, in the present study the downregulation of RORγt among CD3^+^ CD161^+^ lymphocytes in patients with sepsis links to a decrease in Th17 responsiveness and the occurrence of sepsis.

In this study, the proportion of the circulating lymphocyte pool expressing CD3^+^ CD161^+^ was lower in the septic patients who died. This lymphocyte subpopulation is a diverse range of innate and adaptive Th17 cells which may be crucially important in providing some measure of immunity in patients who have an existing adaptive immune paresis [10,12]. This finding suggests that immunity in sepsis is composed of the net effects of several cell populations that together provide innate and adaptive Th17 responses. However, the study was not designed or powered to investigate the mechanism of this association; specifically, whether there was any link between the occurrence of nosocomial infection and the depletion of CD3^+^ CD161^+^ lymphocytes in patients with sepsis.

Innate and adaptive lymphocyte activation was different in patients with sepsis. Many CD3^+^ CD161^+^ lymphocytes are MHC-restricted adaptive immune lymphocytes which were activated in patients with infection but not with sepsis. This contrasts with MAIT, γδ T, and NK cells, which are non-MHC-restricted and were activated in both sepsis and infection. This failure to activate adaptive lymphocytes in septic patients reflects the well-described inhibition of antigen presentation pathways in patients with sepsis and is concordant with recent reports of adaptive immune paresis in septic patients [12].

## 5. Conclusions

Innate lymphocyte depletion was evident in patients with infection and sepsis, suggesting that lymphocyte depletion is not a hallmark of sepsis. Innate lymphocyte phenotypic differences appear to be of greater importance in the pathophysiology of sepsis than the absolute lymphocyte counts. Specifically, sepsis appears to be linked with the downregulation of the innate lymphocyte Th17 phenotype and a failure to upregulate the Th1 phenotype that was evident in patients with infection. In patients with sepsis, in the absence of an adaptive immune response, innate and innate-like T cells appear to assume a crucial role in providing host immunity and may determine the outcome.

A larger clinical study will be required to investigate the utility of specific lymphocyte activation indices as biomarkers for clinical outcome, such as mortality, in infected and septic patients. Lastly, this study demonstrates the benefit of including patients with infection in human studies of sepsis.

## Figures and Tables

**Figure 1 vaccines-08-00311-f001:**
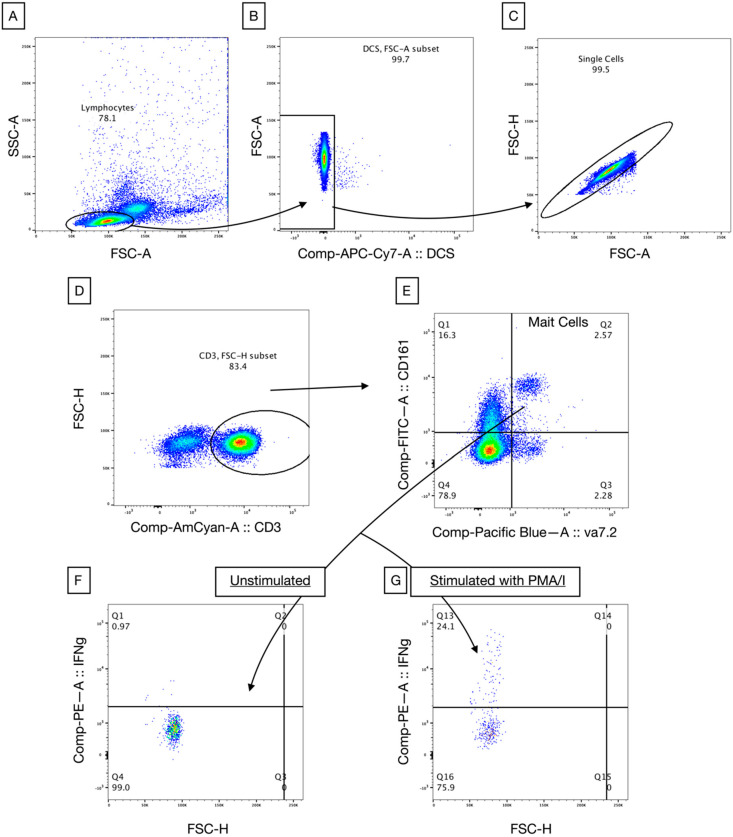
Flow cytometry analysis. Lymphocyte gating on PBMCs (**A**), with exclusion of dead cells (**B**) and exclusion of doublets (**C**). Gating on CD3^+^ cells (**D**) to show MAIT cells (**E**). MAIT cells were gated on showing interferon-γ (IFNγ) expression in unstimulated (**F**) and stimulated (**G**) cells. Stimulation was performed with phorbol myristate acetate and ionomycin (PMA/I). DCS is Dead Cell Stain.

**Figure 2 vaccines-08-00311-f002:**
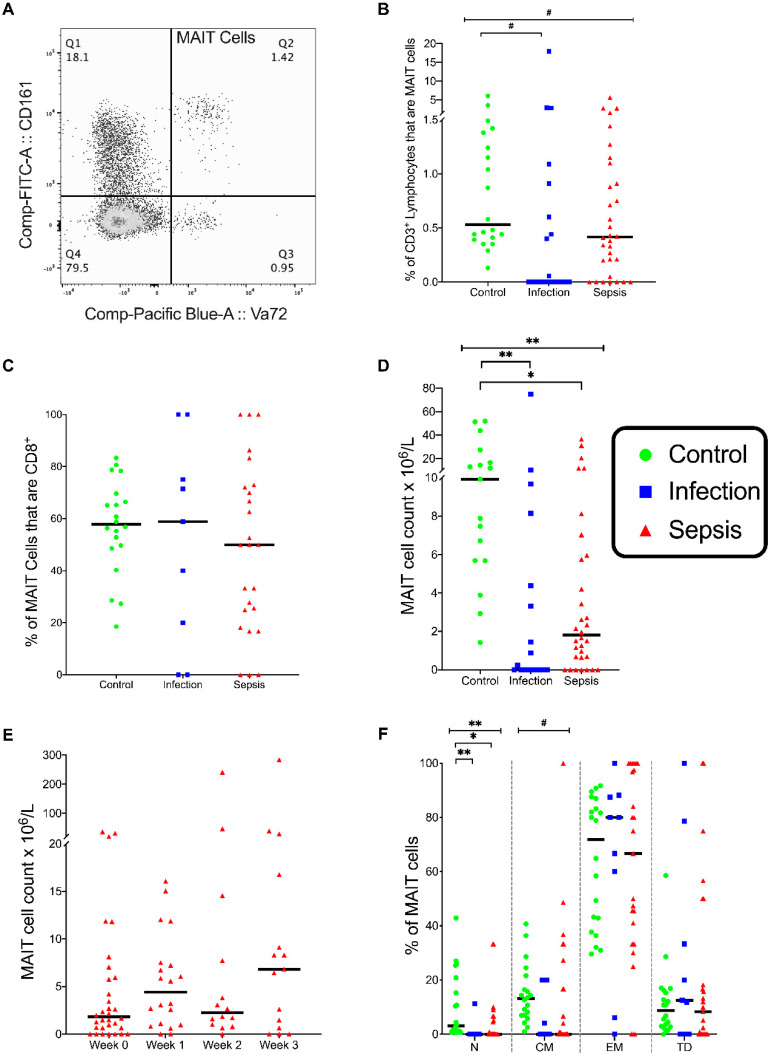
MAIT cells phenotypes. (**A**) Flow cytometry plot showing MAIT cell population (CD3^+^CD161^+^Vα7.2^+^) on gated CD3^+^ lymphocytes. (**B**) Frequency of MAIT cells as a percentage of T cells (CD3^+^ lymphocytes). (**C**) Frequency of MAIT CD8^+^ cells as a % of MAIT cells. (**D**) Total MAIT cell count in circulating blood. (**E**) Total MAIT cell count over time in the septic group. (**F**) Frequencies of naive, central memory, effector memory, and terminally differentiated (N, CM, EM, TD, respectively) MAIT cells in circulating blood. Control group (*n* = 20), infection group (*n* = 19), and septic group (*n* = 32). Graphs are plotted with bars representing the median. ^#^ = *p* < 0.05; * = *p* ≤ 0.01; ** = *p* ≤ 0.001.

**Figure 3 vaccines-08-00311-f003:**
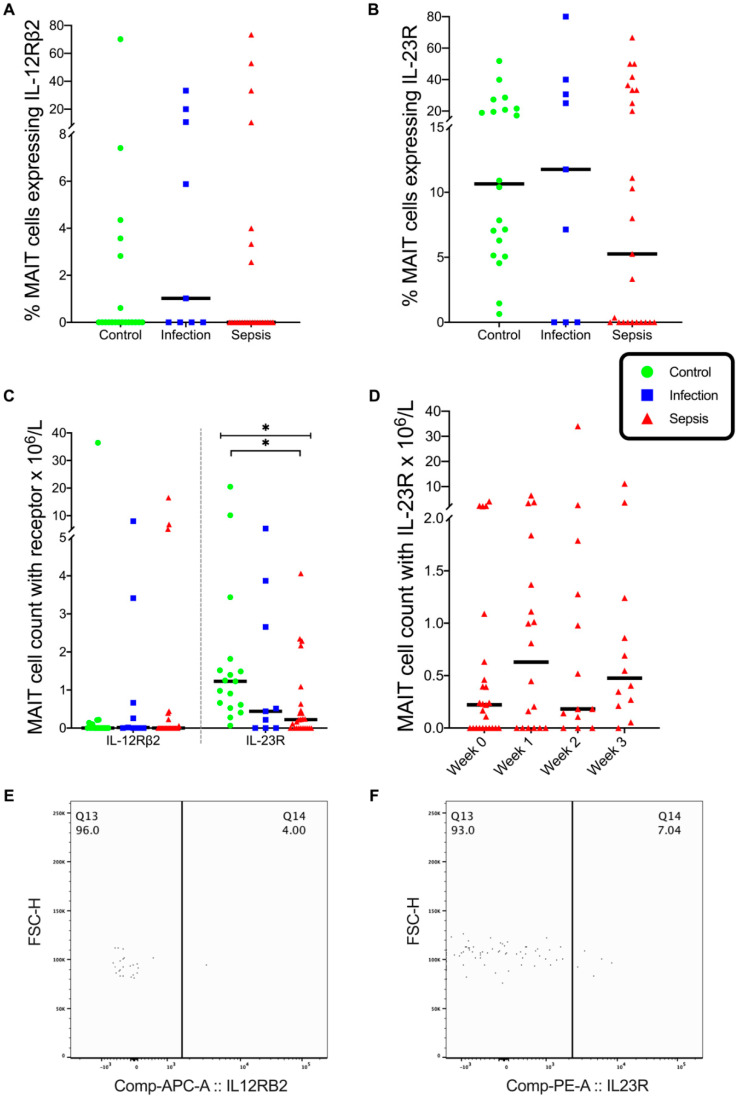
MAIT cell expression of IL-12Rβ2 and IL-23R. (**A**,**B**) Percentage of MAIT cells in circulating blood expressing IL-12 receptor (IL-12Rβ2) and IL-23 receptor (IL-23R). (**C**) MAIT cell count in circulating blood expressing IL-12Rβ2 and IL-23R. (**D**) MAIT cell count expressing IL-23R over time. (**E**) Flow cytometry plot on gated MAIT cells showing IL-12Rβ2. (**F**) Flow cytometry plot on gated MAIT cells showing IL-23 receptor. Control group (*n* = 20), infection group (*n* = 19), and septic group (*n* = 32). Graphs are plotted with bars representing the median. * = *p* ≤ 0.01.

**Figure 4 vaccines-08-00311-f004:**
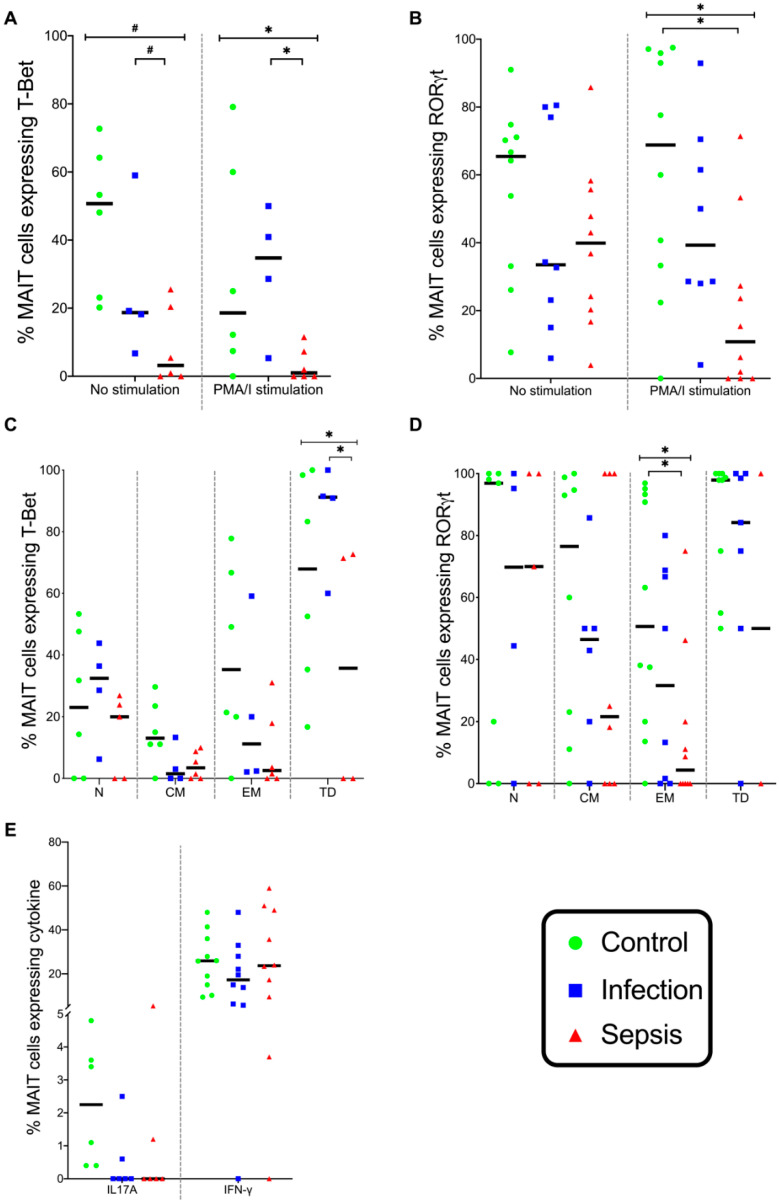
Frequency of RORγt, T-Bet, IL17A, and IFNγ expression in MAIT cells. (**A**) MAIT cell expression of T-Bet when stimulated with the medium alone or PMA/I. (**B**) MAIT cell expression of RORγt when stimulated with the medium alone or PMA/I. (**C**) Expression of transcription factor T-Bet in unstimulated naive, central memory, effector memory, and terminally differentiated (N, CM, EM, TD, respectively) MAIT cells. (**D**) Transcription factor RORγt expression in PMA/I-stimulated N, CM, EM, and TD MAIT cells. (**E**) MAIT cell expression of IL-17a and IFN-γ when stimulated with PMA/I. Control group (*n* = 10), infection group (*n* = 10), and septic group (*n* = 10). Graphs are plotted with bars representing the median. ^#^ = *p* < 0.05; * = *p* ≤ 0.01.

**Figure 5 vaccines-08-00311-f005:**
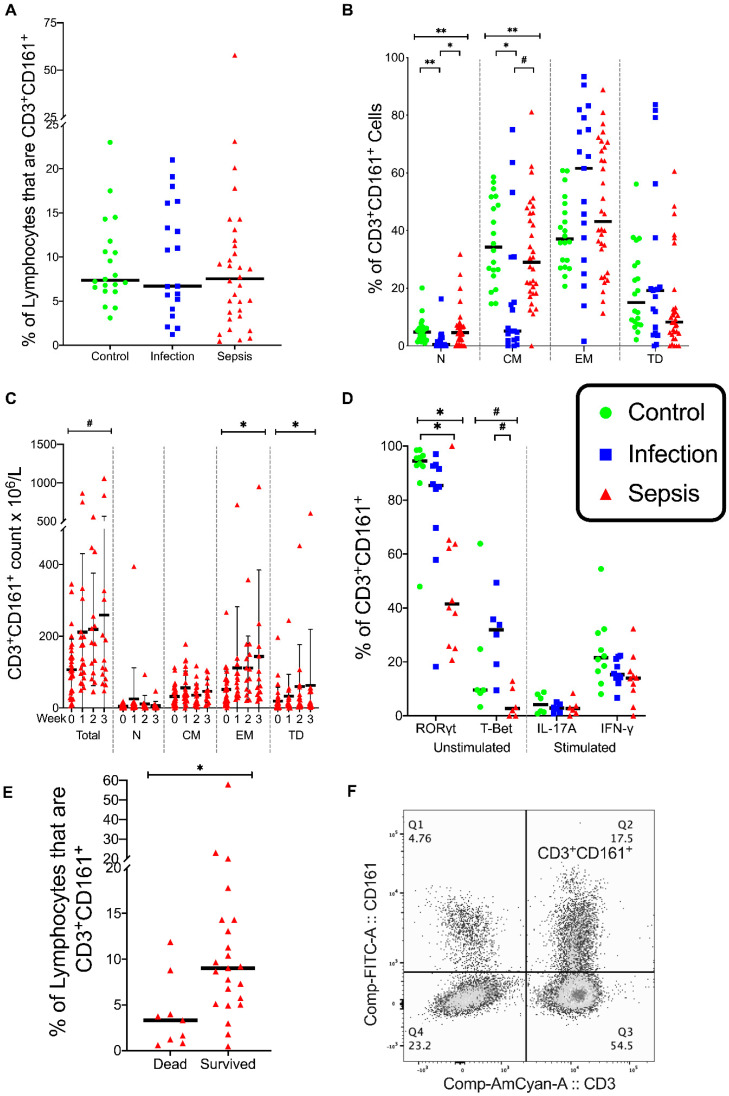
CD3^+^CD161^+^ lymphocytes. (**A**) Frequency of CD3^+^CD161^+^ cells as a percentage of lymphocytes. (**B**) Differentiation status of CD3^+^CD161^+^ lymphocytes in the circulating blood. Naive, central memory, effector memory, and terminally differentiated (N, CM, EM, TD, respectively) differentiation states shown. (**C**) Total number of CD3^+^CD161^+^ lymphocytes and the number of CD3^+^CD161^+^ lymphocytes in their differentiated states in patients with sepsis over time. (**E**) Frequency of lymphocytes that are CD3^+^CD161^+^ in patients who survived compare to those that died. Control group (*n* = 20), infection group (*n* = 19), and septic group (*n* = 32). (**D**) RORγt and T-Bet expression in unstimulated CD161^+^ T cells and cytokines IL17A and IFN-γ in stimulated CD161^+^ T cells with phorbol myristate acetate and ionomycin. Control group (*n* = 10), infection group (*n* = 10), and septic group (*n* = 10). (**F**) Flow cytometry plot of gated lymphocytes showing CD3^+^CD161^+^ cells. (**A**,**B**,**D**,**E**) Graphs are plotted with bars representing the median. (**C**) Data represented as mean with standard deviation. ^#^ = *p* < 0.05; * = *p* ≤ 0.01; ** = *p* ≤ 0.001.

**Figure 6 vaccines-08-00311-f006:**
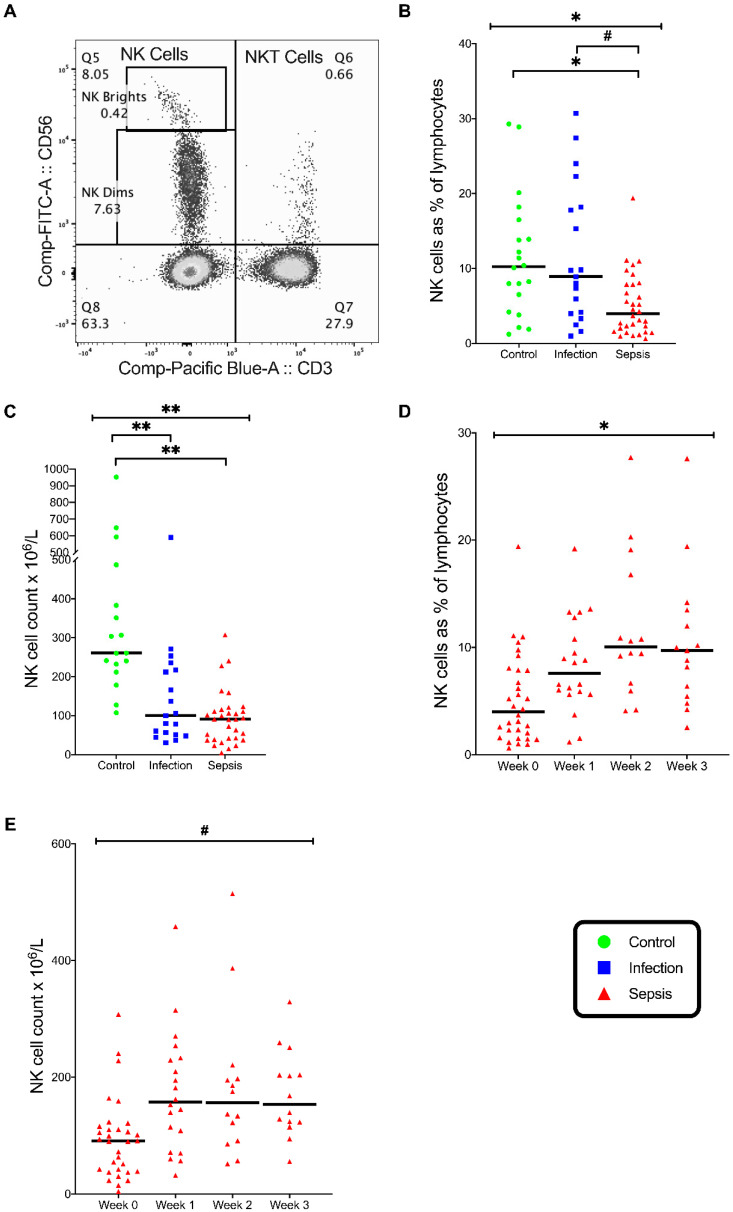
Natural Killer (NK) cells. (**A**) Flow cytometry plot on gated lymphocytes showing NK cells (CD3^−^CD56^+^) and NKT Cells (CD3^+^CD56^+^). (**B**) NK cells as a percentage of lymphocytes. (**C**) NK cell count in the three patient groups on admission. (**D**) Frequency of NK cells as a percentage of lymphocytes in septic patients over time. (**E**) NK cell count in septic patients over time. Control group (*n* = 20), infection group (*n* = 19), and septic group (*n* = 32). Graphs are plotted with bars representing the median. ^#^
*p* < 0.05; * *p* ≤ 0.01; ** *p* ≤ 0.001.

**Figure 7 vaccines-08-00311-f007:**
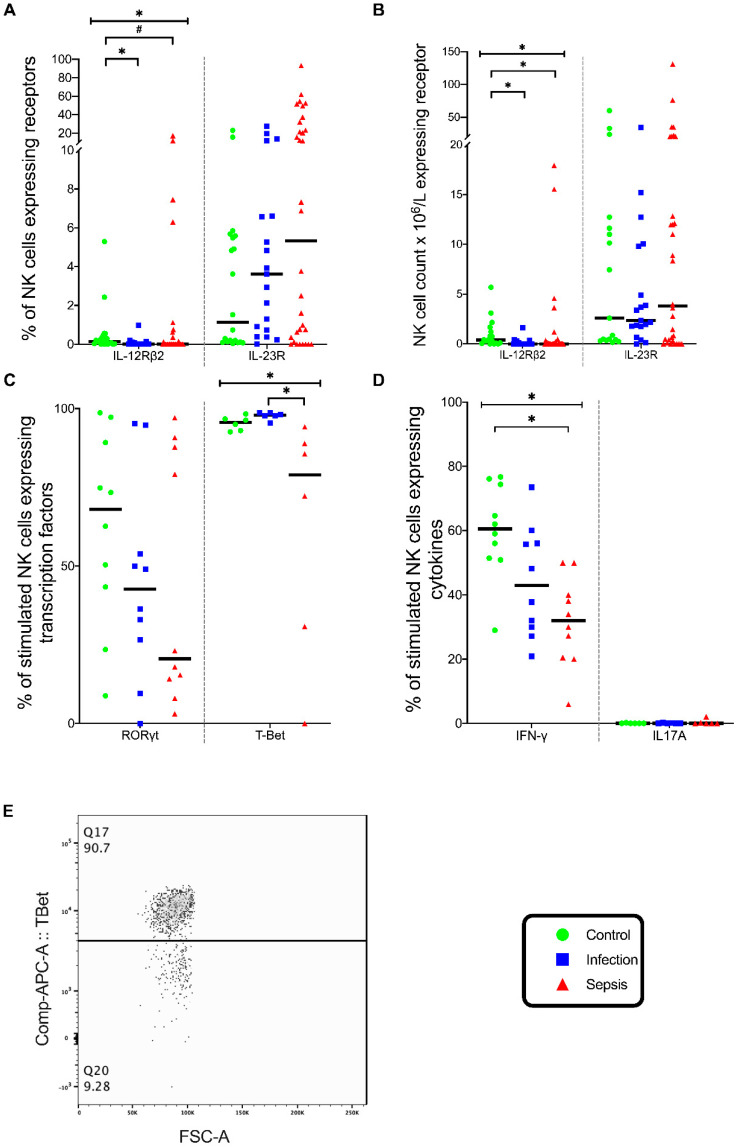
NK cell phenotype and stimulation. (**A**) Frequency of NK cells expressing IL-12 receptor (IL-12Rβ2) and IL-23 receptor (IL-23R). (**B**) Absolute numbers of NK cells expressing IL-12Rβ2 and IL-23R. (**A**,**B**) Control group (*n* = 20), infection group (*n* = 19), and septic group (*n* = 32). (**C**) Frequency of NK cells expressing the transcription factors RORγt and T-Bet when stimulated with phorbol myristate acetate and ionomycin (PMA/I). (**D**) Frequency of NK cells expressing cytokines IL17A and IFN-γ when stimulated with PMA/I. C-D, control group (*n* = 10), infection group (*n* = 10), and septic group (*n* = 10). (**E**) Flow cytometry plot on gated NK cells showing T-Bet expression. All the graphs are plotted with bars representing the median. ^#^ = *p* < 0.05; * = *p* ≤ 0.01.

**Figure 8 vaccines-08-00311-f008:**
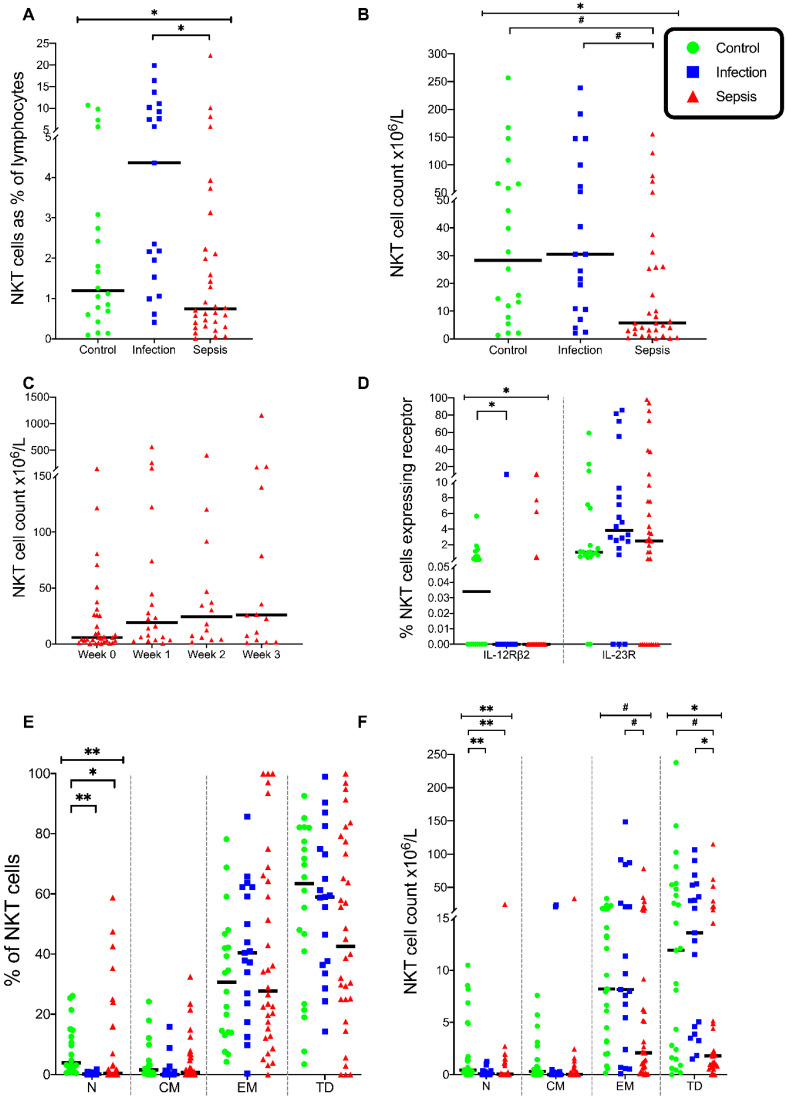
NKT cell phenotype. (**A**) Frequencies of Natural Killer T (NKT) cells as a percentage of lymphocytes. (**B**) Absolute numbers of NKT cells in the three patient groups. (**C**) Total number of NKT cells in septic patients over time. (**D**) Frequency of NKT cells that express the IL-12 receptor (IL-12Rβ2) and IL-23 receptor (IL-23R). (**E**) Differentiation status of NKT cells in circulating blood. Naive, central memory, effector memory, and terminally differentiated (N, CM, EM, TD, respectively) differentiation states shown. (**F**) NKT cell count by differentiation status. Control group (*n* = 20), infection group (*n* = 19), and septic group (*n* = 32). Graphs are plotted with bars representing the median. ^#^ = *p* < 0.05; * = *p* ≤ 0.01; ** = *p* ≤ 0.001.

**Figure 9 vaccines-08-00311-f009:**
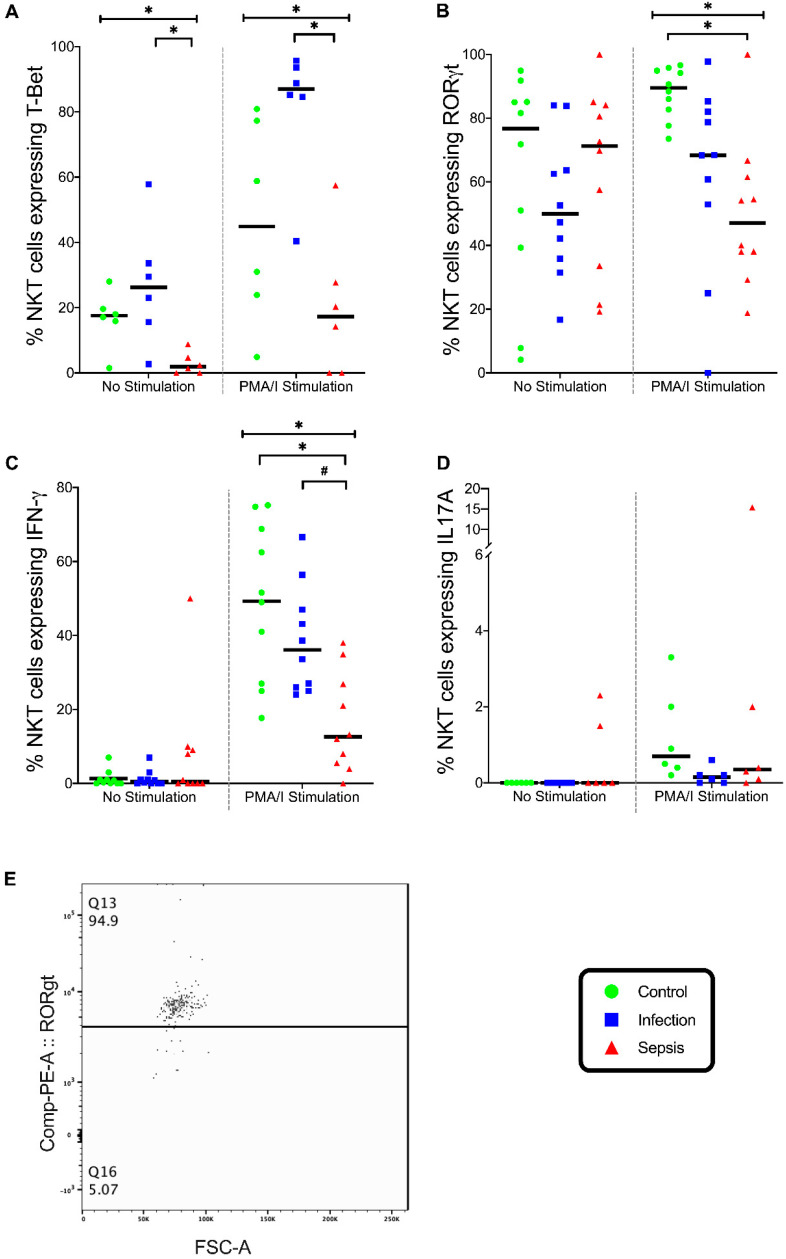
NKT cell stimulation. (**A**) Frequencies of unstimulated Natural Killer T (NKT) cells and stimulated NKT cells with phorbol myristate acetate and ionomycin (PMA/I) expressing T-Bet. (**B**) Frequency of unstimulated NKT cells and stimulated NKT cells with PMA/I expressing RORγt. (**C**) Frequency of unstimulated NKT cells and stimulated NKT cells with PMA/I expressing IFN-γ. (**D**) Frequency of unstimulated NKT cells and stimulated NKT cells with PMA/I expressing IL17A. (**E**) Flow cytometry plot on gated NKT Cells expressing RORγt. Control group (*n* = 10), infection group (*n* = 10), and septic group (*n* = 10). Graphs are plotted with bars representing the median. ^#^ = *p* < 0.05; * = *p* ≤ 0.01.

**Figure 10 vaccines-08-00311-f010:**
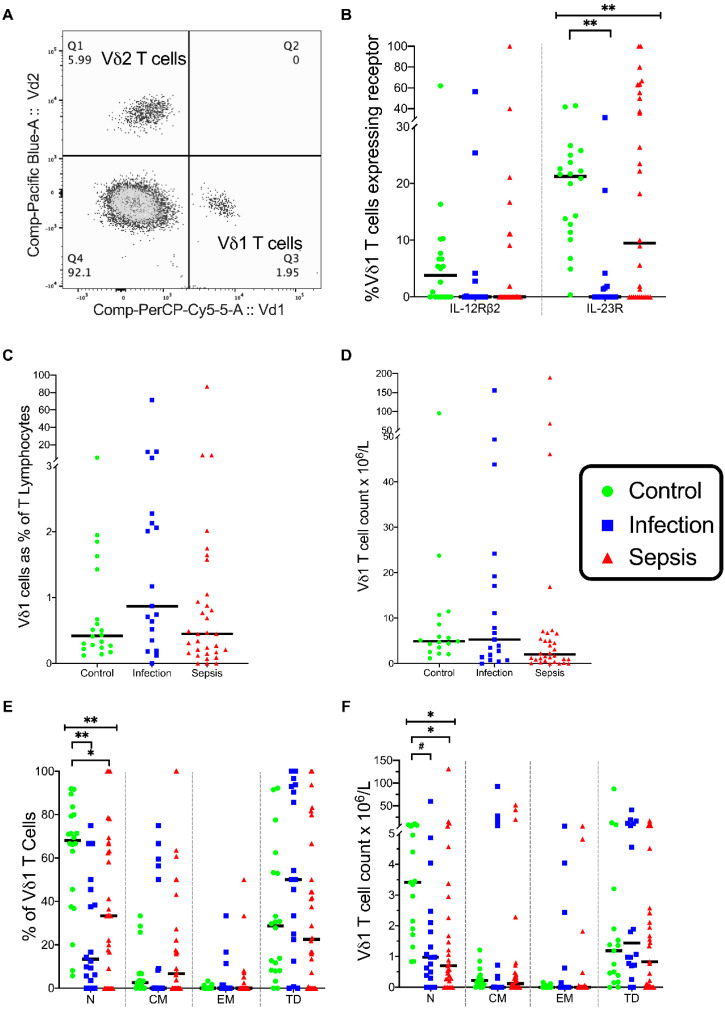
Vδ1 cell phenotype. (**A**), Flow cytometry plot on gated T Cells showing Vδ1 and Vδ2 cells. (**B**) Frequency of Vδ1 T cells expressing IL-12 receptor (IL-12Rβ2) and IL-23 receptor (IL-23R). (**C**) Frequency of T cells that express Vδ1 TCRs. (**D**) Vδ1 T cell count in the three patient groups. (**E**) Differentiation status of Vδ1 T cells. Naive, central memory, effector memory, and terminally differentiated (N, CM, EM, TD, respectively) differentiation states shown. (**F**) Differentiation status of Vδ1 T cells expressed as the total cell count. Control group (*n* = 20), infection group (*n* = 19), and septic group (*n* = 32). Graphs are plotted with bars representing the median. ^#^ = *p* < 0.05; * = *p* ≤ 0.01; ** = *p* ≤ 0.001.

**Figure 11 vaccines-08-00311-f011:**
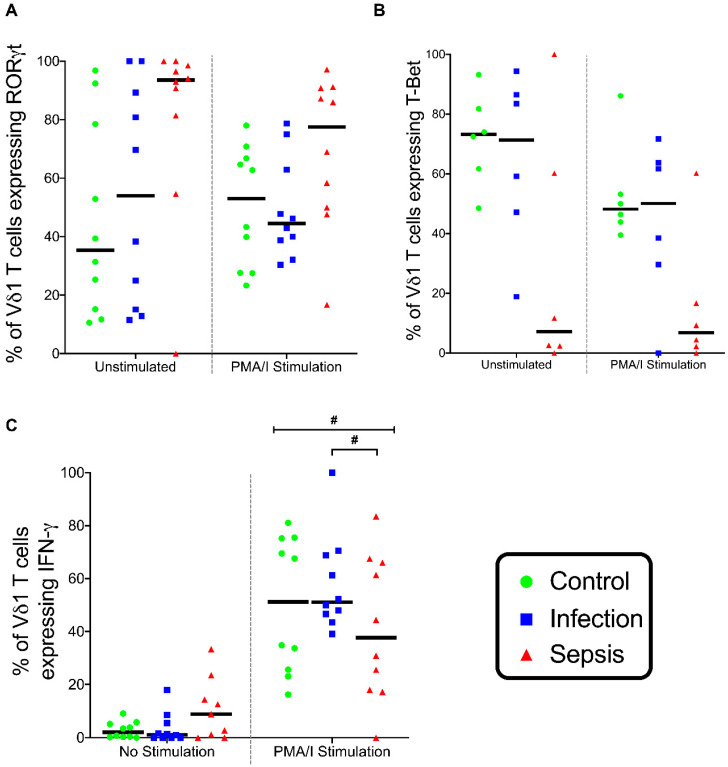
Vδ1 cell stimulation. (**A**) Frequency of unstimulated and stimulated Vδ1 cells expressing RORγt. (**B**) Frequency of unstimulated and stimulated Vδ1 cells expressing T-Bet. (**C**) Frequency of unstimulated and stimulated Vδ1 cells expressing IFN-γ. All were stimulated with phorbol myristate acetate and ionomycin (PMA/I). Control group (*n* = 10), infection group (*n* = 10), and septic group (*n* = 10). Graphs are plotted with bars representing the median. ^#^ = *p* < 0.05.

**Figure 12 vaccines-08-00311-f012:**
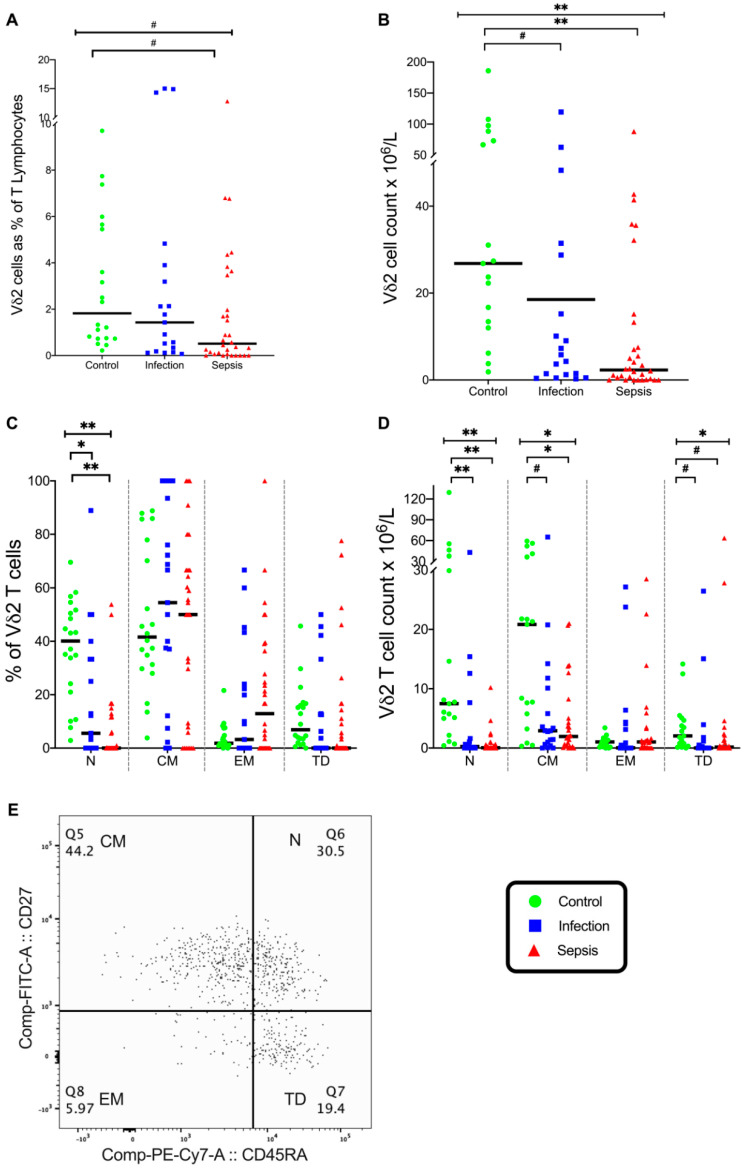
Vδ2 cell phenotype. (**A**) Frequency of T lymphocytes that are Vδ2 Cells. (**B**) Vδ2 T cell count in the three patient groups. (**C**) Differentiation status of Vδ2 T cells. Naive, central memory, effector memory, and terminally differentiated (N, CM, EM, TD, respectively) differentiation states shown. (**D**) Differentiation status of Vδ2 T cells expressed as the total cell count. (**E**) Flow cytometry plot on gated Vδ2 T cells showing differentiation status. Control group (*n* = 20), infection group (*n* = 19), and septic group (*n* = 32). Graphs are plotted with bars representing the median. ^#^ = *p* < 0.05; * = *p* ≤ 0.01; ** = *p* ≤ 0.001.

**Figure 13 vaccines-08-00311-f013:**
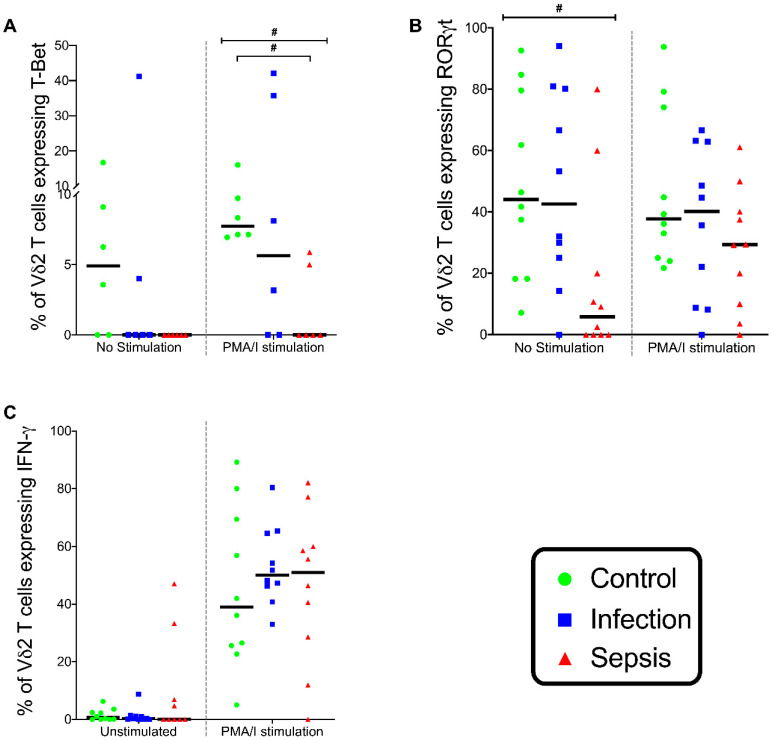
Vδ2 cell stimulation. (**A**) Frequency of unstimulated and stimulated Vδ2 T cells expressing T-Bet. (**B**) Frequency of unstimulated and stimulated Vδ2 T cells expressing RORγt. (**C**) Frequency of unstimulated and stimulated Vδ2 T cells expressing interferon gamma (IFN-γ). Stimulated with phorbol myristate acetate and ionomycin (PMA/I). Control group (*n* = 10), infection group (*n* = 10), and septic group (*n* = 10). Graphs are plotted with bars representing the median. ^#^ = *p* < 0.05.

**Table 1 vaccines-08-00311-t001:** Demographic and clinical data.

Clinical Data		Immunophenotype Study			Stimulation Experiment		
		Control	Infection	Septic	Control	Infection	Septic
*n*		20	19	32	10	10	10
Age		73 [69–78.25]	81.5 [70.25–87.25]	73.5 [68.75–79.25]	72 [67–74.5]	85 [68.75–87.5]	57.5 [53.75–69.25]
Male Gender		6 (30%)	11 (58%)	20 (62.5%)	5 (50%)	6 (60%)	4 (40%)
APACHE score		N/A	12.5 [8–16.5]	19 [16–24.5] *p* < 0.0001	N/A	14 [10–14]	21.5 [16.25–24.5]
SAPS score		N/A	N/A	48 [37.75–54.5]	N/A	N/A	49 [38.25–55.75]
SOFA score on admission		N/A	3 [1.75–4]	7 [5.75–10] *p* < 0.0001	N/A	2 [0–3]	10 [7.5–11.75]
SOFA score on day of first sample		N/A	1 [0.75–1.25]	7 [5–8.25]	N/A	1 [0,1]	8 [4.5–9.75]
Time to 1st sample from admission (days)		N/A	2.5 [2,3]	1.5 [0.75–2]	N/A	3 [2.5–4.5]	5 [4–6]
ICU duration (days)		N/A	N/A	14.5 [8.75–33.25]	N/A	N/A	17.5 [9.25–26]
Mortality in ICU		N/A	N/A	11 (34.4%)	N/A	N/A	1 (10%)
Mortality in Hospital		N/A	0	13 (40.6%)	N/A	1 (10%)	3 (30%)
Inotropic Support		N/A	0	30 (93.75%)	N/A	0	10 (100%)
Days on inotropes		N/A	0	7 [3–13]	N/A	0	7.5 [6–10.5]
Invasive ventilation		N/A	N/A	28 (87.5%)	N/A	N/A	9 (90%)
Days on invasive ventilation		N/A	N/A	14.5 [5–29.25]	N/A	N/A	8.5 [6.25–14.5]
*p*/F ratio (mmHg)		N/A	265.5 [331–392.25]	170 [135.75–240.5]	N/A	411.5 [386.5–436.5]	163 [129–205.75]
Muscle Relaxant infusion		N/A	N/A	11 (34.4%)	N/A	N/A	5 (50%)
Acute Kidney Injury KDIGO grade ≥1		0	6 (31.58%)	26 (81.25%)	0	2 (20%)	10 (100%)
Renal Replacement Therapy		0	0	16 (50%)	0	0	8 (80%)
Stress dose steroids		0	1 (5.3%)	6 (18.75%)	0	1 (10%)	5 (50%)
Source of Sepsis	Respiratory	N/A	5 (26.3%)	16 (50%)	N/A	3 (30%)	4 (40%)
	Abdominal	N/A	7 (36.8%)	11 (34.4%)	N/A	2 (20%)	3 (30%)
	Skin	N/A	0	4 (12.5%)	N/A	1 (10%)	1 (10%)
	Urine	N/A	6 (31.6%)	0	N/A	4 (40%)	2 (20%)
	Osteomyelitis	N/A	1 (5.3%)	0	N/A	0	0
	Mediastinitis	N/A	0	1 (3.1%)	N/A	0	0
Type of organism	Gram− ve organism	N/A	16 (84%)	9 (28%)	N/A	5 (50%)	5 (50%)
	Gram+ ve organism	N/A	3 (16%)	9 (28%)	N/A	5 (50%)	2 (20%)
	Fungal	N/A	0	1 (3%)	N/A	0	0
	Empiric treatment	N/A	0	13 (41%)	N/A	0	3 (30%)
Secondary Infections		N/A	0	17 (53.1%)	N/A	0	5 (50%)
Lactate on admission		N/A	2.1 [1.15–3.57]	2.86 [2.27–4.09]	N/A	1.95 [1.6–3.1]	2.63 [1.81–3.36]
Comorbidities ≥ 1		19 (95%)	18 (94.7%)	26 (81.25%)	7 (70%)	9 (90%)	8 (80%)

Categorical data are presented as numbers with percentages in parentheses (). Data and parameters are presented as medians and interquartile ranges [Q1–Q3]. n, number of patients; APACHE score, Acute Physiologic Assessment and Chronic Health Evaluation; SOFA score, Sequential Organ Failure Assessment score; KDIGO Kidney Disease: Improving Global Outcomes; N/A, Not Applicable; *p*/F ratio, ratio of arterial oxygen partial pressure to fractional inspired oxygen.

**Table 2 vaccines-08-00311-t002:** Exclusion criteria.

	Exclusion Criteria
1	Any haematological disease
2	Pre-existing liver disease
3	Pre-existing immunodeficiency
4	Immune modulating medications (incl. steroids prior to onset of sepsis)
5	Chronic infection
6	Malignancy

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
