# Peer review of "Innate Lymphocyte Th1 and Th17 Responses in Elderly Hospitalised Patients with Infection and Sepsis"

_vaccines, 2020, doi:10.3390/vaccines8020311_

Round 1

Reviewer 1 Report

The MS presents interesting clinical study of sepsis patients using mainly flow cytometry based approach and does provide useful information about the cell phenotype in elderly septic patients.

The study is conducted in methodical way and data is presented well but in a complicated way for common readers. The detail clinical table is appreciated. stress dose steroids in the table but as per exclusion criteria steroid were excluded?

Though the results are interesting but huge error bars are a concern and data may be verified by expert statistician. 

There should be a clear statement about the first time novel findings coming out from this study highlighting the implication of the results shown. A conclusion part may be added.

It will be much better to include more recent references and comparison of results in the discussion part.

Author Response

The MS presents interesting clinical study of sepsis patients using mainly flow cytometry based approach and does provide useful information about the cell phenotype in elderly septic patients.

The study is conducted in methodical way and data is presented well but in a complicated way for common readers. The detail clinical table is appreciated. stress dose steroids in the table but as per exclusion criteria steroid were excluded?

In the clinical table “stress dose steroids” in the table refers to physiological steroids given to a patient in profound septic shock on  high doses of vasopressors after the diagnosis of sepsis was made. The exclusion criteria refers to patients receiving any steroids prior to sepsis being diagnosed. I have updated the exclusion table to reflect this.

Though the results are interesting but huge error bars are a concern and data may be verified by expert statistician. 

I have reviewed all data submitted to the statistician, this data can be found on private link: https://figshare.com/s/01aae3823a0baf3c90cb. This data will become public open access once the paper is published (DOI to become active when paper published: 10.6084/m9.figshare.12229346). On reviewing all  the data submitted I noted that the RORγt and T-Bet data for the Vδ1 and Vδ2 Cells was not correct and I apologise for this. All other data was correct. The graphs presented initially were median with IQR. These graphs have been updated to individual values with a median bar so that all values can be seen on the graph, this was done on suggestion of the other reviewer. The statistics have been reverified and the same method of statistical analysis was used in our previous published paper (reference 12 of manuscript). The statistical analysis was planned in advance of the commencement of the study. During the reverification of the statistics we have corrected the RORγt and T-Bet results for the Vδ1 and Vδ2 Cells and have updated the manuscript accordingly.

There should be a clear statement about the first time novel findings coming out from this study highlighting the implication of the results shown. A conclusion part may be added.It will be much better to include more recent references and comparison of results in the discussion part.

The discussion has been updated and a conclusion added. More references have been added.

Reviewer 2 Report

The manuscript by Oakley et al describes response of different lymphocytes in elderly hospitalized patients with infection and sepsis. The topic is of interest and study is well designed. The problem is data collection and presentation. There was no primary data provided and it is hard to confirm their frequencies looking at bar graph. In some cases, the figures are not labeled properly. In some cases, correct references are not cited. As such manuscript is hard to follow for example the abstract discuss about MAIT cells and Figure 1 is the gating strategy of NK cells. This study tries to establish links of the mosaic of innate lymphocyte phenotypes with the clinical presentation of infection and sepsis in patients. Looking at the current manuscript. Clinically it is not clear why different infections and sepsis all lead to a decrease in lymphocyte numbers.  Could this all be due to fever or result of some pyrogen?

  1. The abstract is hard to follow. It is not clear on the rationale and even results.

  1. “In the presence of such a profound paresis of adaptive immunity, the function of innate lymphocytes in the pathophysiology of human sepsis should be clarified, in order to identify potential avenues for novel immune adjuvant therapies, but also to provide an enhanced suite of biomarkers of immunity in human sepsis.” I can’t understand what author are trying to convey here.

  1. “Innate lymphocytes, such as MAIT cells, bridge the gap between innate and adaptive immunity 75 [13].” MAIT cells are a subset of T cells that show innate-like effector response are they truly innate cells.
  2. “MAIT cells are the most numerous invariant T lymphocytes in the peripheral circulation and are also found in the gastrointestinal and respiratory tracts” This reference does not show homing to blood but intestine and liver.
  3. “Both NK cells and NKT cells are important sources of early IFN-γ but can also be induced to produce other cytokines like IL-17 [17,18] and these cells are variably reported as reduced or increased in patients with sepsis [19,20].” There is generally an issue with wrong references in this paper. These references are all on NKT cells and NK. Further, I cant see they confirm NK or NKT are primary sources of early IL-17 in these references.

  1. Figure 1 gating strategy is not clear. The lymphocytes population seems not right for FSC/SSC gate (too low FSC/SSC). Did you check the other population? Your doublet gate is incorrect. The live dead stain shows almost no dead cells. Your NK population is 30% of Total and T-cells are too low. I will like to see the live dead staining/antibody stain in ungated and different populations.

  1. Since this paper is more about MAIT cells I am surprised authors did not show the staining strategy of MAIT cells.

  1. One of the major issues of this manuscript is the lack of primary flow data. The authors should include the representative Flow cytometry profile of different experiments.

  1. Figure 2A What is on Y-axis.
  2. Please change all histograms to graph type where individual data points and variation can be seen.

  1. In Figure 2, The frequency of MAIT cells can change due to variation in the overall T-cell number. Did the authors check the frequency of T-cells in PBMCs?
  2. “Frequencies of Naive, Central Memory, Effector Memory, and Terminally Differentiated (N,CM,EM,TD respectively) MAIT cells in circulating blood. “ It is difficult to interpret your data without looking at primary flow data.

  1. Figure 3A change your Y-axis I cant see the bar.
  2. Figure 5: why authors show standard error of the mean of data which should use standard deviation since it comes from different subjects. The authors should check statistics and apply appropriate tests.

  1. Figure 7 C and D The Y-axis is the same and does not say expressing what?

Author Response

Please find reply attached. 

Round 2

Reviewer 1 Report

Authors have addressed my comments in revised MS.

Author Response

Open Review

English language and style

( ) Extensive editing of English language and style required  
( ) Moderate English changes required  
(x) English language and style are fine/minor spell check required  
( ) I don't feel qualified to judge about the English language and style  

Yes

Can be improved

Must be improved

Not applicable

Does the introduction provide sufficient background and include all relevant references?

(x)

( )

( )

( )

Is the research design appropriate?

(x)

( )

( )

( )

Are the methods adequately described?

(x)

( )

( )

( )

Are the results clearly presented?

( )

(x)

( )

( )

Are the conclusions supported by the results?

( )

(x)

( )

( )

Comments and Suggestions for Authors

Authors have addressed my comments in revised MS.

Submission Date

03 May 2020

Date of this review

01 Jun 2020 16:11:03

A change I have made to the text is to refer to gamma-delta T Cells as “γδ T Cells” rather than Vδ T Cells but i have left the gamma-delta T Cells subgroups referred to as “Vδ1” and “Vδ2.”

The flow cytometry plot in Figure 3 gating for IL12RB2 (Fig 3E) has been changed to have a more comparable FSC-H to Figure 3F at the request of another reviewer.

An addition to the text (line 250) in the demographic section as requested by another reviewer is as follows:

 “There were more male than female patients in the immunophenotyping study. This is consistent with sepsis being a disorder of the elderly and then primarily elderly males, as there are twice as many males with sepsis than females.”

I have inserted an additional reference for this (reference 29).

Many thanks for reviewing this paper.

Reviewer 2 Report

Overall authors have addressed all my primary comments and revised manuscript. I have few additional comments to help authors before acceptance of this manuscript.

  1. “The three groups had similar gender and age demographics” Authors suggest the three groups have similar gender while in immunophenotype study shown in Table 1 it is biased for Male. These results should be taken with caution and this should be mentioned in abstract and discussion.

2. There are several details missing in Table 1 for example the ICU duration is only provided for Sepsis but not for infected group.

3. During Infection MAIT cell number go down drastically in blood which is different from sepsis. In the figure 2c with MAIT cells that are CD8+ the sample number goes down is this the reason the three groups looks similar.

4. In Figure 3E the cell population looks weird on Forward side scatter. It is on 50k compared to 100k almost every other time. Please check the gating strategy.

Author Response

Open Review

English language and style

( ) Extensive editing of English language and style required  
( ) Moderate English changes required  
(x) English language and style are fine/minor spell check required  
( ) I don't feel qualified to judge about the English language and style  

Comments and Suggestions for Authors

Overall authors have addressed all my primary comments and revised manuscript. I have few additional comments to help authors before acceptance of this manuscript.

  1. “The three groups had similar gender and age demographics” Authors suggest the three groups have similar gender while in immunophenotype study shown in Table 1 it is biased for Male. These results should be taken with caution and this should be mentioned in abstract and discussion.
  2. There are several details missing in Table 1 for example the ICU duration is only provided for Sepsis but not for infected group.
  3. During Infection MAIT cell number go down drastically in blood which is different from sepsis. In the figure 2c with MAIT cells that are CD8+ the sample number goes down is this the reason the three groups looks similar.
  4.  In Figure 3E the cell population looks weird on Forward side scatter. It is on 50k compared to 100k almost every other time. Please check the gating strategy.

Submission Date

03 May 2020

Date of this review

10 Jun 2020 21:34:15

Reply:

1. In the results section in Demographics (line 249) I have altered that sentence as follows:

“The three groups had similar age demographics. There were more male than female patients in the immunophenotyping study. This is consistent with sepsis being a disorder of the elderly and then more so in males.” An additional reference for this statement is added [29].

2. In Table 1 there are several data boxes in the control and infection groups with “Not Applicable” in them. Regarding the Control group: these were subjects who were never in the hospital and were healthy controls from the community and so many sections are not applicable to them. Regarding the Infection group these are patients recruited from the general wards in the hospital as detailed in methods and materials. Thus they were never admitted to the Intensive Care Unit as they did not develop the need for organ support. This is why there are sections in the table that are not applicable to them as they were not in icu needing support. This makes this population a very interesting group to compare to the sepsis group as clearly although they had an infection their immune system was able to cope with the infection and not result in organ failure.

3. Indeed the MAIT cell frequency and numbers go down significantly when compared to control. The MAIT cell count reaches a lower p value in the infection group than the sepsis group when compared to the control group. However there is no statistical significance when comparing the infection group with the sepsis group. This is most likely due to the low numbers of MAIT cells found in both the infection and septic group. In the discussion we note that a low MAIT cell count in sepsis has been seen before (Grimaldi et al). We suggest that numeric depletion of MAIT cells is a common feature of all infections, with or without sepsis, and may reflect redistribution rather than depletion.

Regarding Figure 2C, the sample number goes down as you have noted, more so in the infection group. This is because when a sample had Zero MAIT cells then it is impossible to have a % of zero MAIT cells expressing CD8. There were more samples in the infection group with no MAIT cells found. Of the MAIT cells that were present in the circulating blood in all groups the frequency of MAIT cells that were CD8+ was similar. 

4. In Figure 3 the FSC-H values differ in parts E and F because the samples were taken from different subjects.  We found that the FSC-H values differed in different subjects, especially when comparing patients and controls. We ensured that it was the lymphocyte population being gated on in each sample. We have now replaced this flow cytometry plot in Figure 3E with one with an FSC-H from another subject that is more comparable with that in Figure 3F

A change I have made to the text is to refer to gamma-delta T Cells as “γδ T Cells” rather than Vδ T Cells but i have left the gamma-delta T Cells subsets referred to as “Vδ1” and “Vδ2.”

Many thanks for reviewing this paper.